# Review of IoT Sensor Systems Used for Monitoring the Road Infrastructure

**DOI:** 10.3390/s23094469

**Published:** 2023-05-04

**Authors:** Kristian Micko, Peter Papcun, Iveta Zolotova

**Affiliations:** Department of Cybernetics and Artificial Intelligence, Faculty of Electrical Engineering and Informatics, Technical University of Kosice, 042 00 Kosice, Slovakia; kristian.micko@tuke.sk (K.M.); iveta.zolotova@tuke.sk (I.Z.)

**Keywords:** edge computing, transportation tasks, motion detection, object tracking, object detection, object classification, intrusive sensors, non-intrusive sensors, IoT sensor systems, monitoring the road infrastructure

## Abstract

An intelligent transportation system is one of the fundamental goals of the smart city concept. The Internet of Things (IoT) concept is a basic instrument to digitalize and automatize the process in the intelligent transportation system. Digitalization via the IoT concept enables the automatic collection of data usable for management in the transportation system. The IoT concept includes a system of sensors, actuators, control units and computational distribution among the edge, fog and cloud layers. The study proposes a taxonomy of sensors used for monitoring tasks based on motion detection and object tracking in intelligent transportation system tasks. The sensor’s taxonomy helps to categorize the sensors based on working principles, installation or maintenance methods and other categories. The sensor’s categorization enables us to compare the effectiveness of each sensor’s system. Monitoring tasks are analyzed, categorized, and solved in intelligent transportation systems based on a literature review and focusing on motion detection and object tracking methods. A literature survey of sensor systems used for monitoring tasks in the intelligent transportation system was performed according to sensor and monitoring task categorization. In this review, we analyzed the achieved results to measure, sense, or classify events in intelligent transportation system monitoring tasks. The review conclusions were used to propose an architecture of the universal sensor system for common monitoring tasks based on motion detection and object tracking methods in intelligent transportation tasks. The proposed architecture was built and tested for the first experimental results in the case study scenario. Finally, we propose methods that could significantly improve the results in the following research.

## 1. Introduction

Transportation systems are the backbone of every economy in the world. This study will focus on the Internet of Things (IoT) concepts used in road transportation infrastructure. IoT concepts applied to road transportation infrastructure can create the part of an intelligent transportation system. An intelligent transportation system could increase traffic efficiency and safety because the road’s monitoring system would be reliable. The automatization of monitoring systems can be performed in different ways, and this article compares the advantages and disadvantages of the most commonly used ways.

Organizing fluent and safe traffic is a priority in every country. By defining monitoring tasks in intelligent transportation systems, it is necessary to achieve this goal and characterize the data to be collected. In compliance with [1], there are measured data about safety, the diagnostics of a vehicle or infrastructure, traffic management, driver’s assistance, environment’s impact and user’s ability to drive. We focused on a comparison of the sensor systems collecting traffic management and safety data. The sensor systems used for traffic management and safety data use the techniques for vehicle motion detection and object tracking. The traffic and safety management system requires monitoring tasks such as vehicle type classification and counting, automatic license plate recognition, incident detection, parking management and speed measurement. Regarding the opinion of the authors in the scientific article [2], many sensor systems have been developed to solve these tasks, but their effectiveness has not significantly improved over the past decade. We propose a research direction where using the IoT concept via computational distribution on edge computing in sensor systems can improve the sensor system’s effectiveness.

This paper is organized as follows. The IoT concept is introduced in Section 2. The IoT concept includes several main parts, such as sensors, control units, actuators and computational distribution. We focus on computational distribution and propose our taxonomy according to its purpose in transportation tasks as well as a taxonomy from the view of heterogeneity sensor data. Section 3 briefly explains the main working principles of sensors used for intelligent transportation tasks. Section 4 describes transportation tasks based on motion detection and object tracking. Section 5 compares sensor systems’ effectiveness, as mentioned in Section 3, in collecting data for the tasks mentioned in Section 4 based on a state-of-the-art literature survey. We rely on the comparison results in Section 5 to select the sensors for our architecture of the universal sensor system in Section 6. Section 7 presents a case study scenario where the proposed architecture of a universal sensor system is described, as well as its implementation and the first experimental results. We discuss research challenges and gaps in Section 8 and conclude them in Section 9.

## 2. IoT Concepts

The IoT is a concept of connecting devices to the internet. This connection helps to create a digital copy of the physical world. The main attributes of the IoT concept are sensors, actuators, control units, and the internet connecting these entities. The Internet is an instrument that connects each part of the IoT concept, and actuators are devices used to execute actions in the IoT concept. In the opinion of the authors [3,4], the IoT concept used for intelligent transportation systems is an important step in improving the quality of the transportation industry. In this section, we introduce the taxonomy of sensors and the possibility of computation distribution on edge, fog, and cloud computing.

### 2.1. Taxonomy of Sensors

Further, a sensor is defined as an entity that retrieves the state of a measured or sensed object and sends the data or information to a control entity [5]. We can use living organisms [6] or some devices [7] as sensors.

We propose the categorization sensors according to the taxonomy mentioned in Figure 1. Sensors could be divided into physical or virtual ones. Physical sensors are devices directly measuring the state of sensed things, and the particular state is being send to a control or storage unit [8]. Virtual sensors are entities that indirectly measure an object’s state via computation, aggregation, or other processing methods from gathered data [9,10].

Virtual sensors can be divided into three categories [11,12]:Software-based;Hardware-based;A combination of software- and hardware-based.

Software-based virtual sensors are devices with a deployed program that senses the state of an object in a virtual world. For example, it describes the model of a neural network that is trained for sentiment analysis used to measure aggressiveness behaviour in forum conversation. This model changes the computer into a social behaviour sensor [13,14,15].

The hardware-based virtual sensor shows the object’s measured value from the computed value of two or more physical sensors connected to a computer or a microcontroller unit (MCU). For instance, temperature and humidity sensors connected to MCU, fuse data using the computation of atmospheric pressure in the observed environment. The MCU computes the pressure value and behaves like a virtual barometer [12].

Virtual sensors, made with fusion software- and hardware-based approach, are physical sensors connected to a control unit collecting data from the other virtual sensor to compute the measured objects [11]. For example, two cameras of the same type could be used for stereovision purposes. The original purpose of these cameras was to take pictures or videos. The algorithms for chess calibration and matching pixels’ positions in pictures taken by both cameras, based on the fact that the positions of each camera are known, are used for distance calculation of the objects of interest from the taken pictures. The software using the algorithm to calculate the distance of each pixel from taken pictures is a virtual sensor for distance measurement. A virtual sensor transforms pixels to voxels. The cameras are physical sensors that provide the data for calculation.

The physical sensors mentioned in this paper are divided into three main categories [16,17]:mechanical;electronic;biological.

The mechanical sensor we describe is a sensor that measures the value of an object’s state in the mechanical meter; for example, the volume of the mercury in the glass cylinder measures temperature [18].

An electronic sensor senses the object’s state by converting physical quantities into electrical signals; for example, piezoelectric sensors make electric impulses when they are impacted by pressure deformation [19].

There is correlation to the fact that a biological sensor is a living organism that shows the status of the measured environment’s state by changing its behaviour, such as Syngonium Podophyllum becoming fluorescent in soil that contains Arsenicum or other heavy metals [6].

### 2.2. Edge and Fog Computing

There are many definitions of edge and fog computing. Some authors consider these words synonymous, but we distinguish between the edge and fog computing paradigms (Figure 2). Both paradigms are based on computing in the nearest node to the data source, but they can be used for other purposes [20,21,22].

Edge computing runs on edge devices such as smartphones, single-board computers (SBC), or MCUs directly connected to the sensors or actuators. Within them there are implemented programs that process the data received from sensors that are sent to the central computational node [23] or directly to the actuator. It is possible to convert the edge device to a virtual sensor. The edge device with implemented software can sense other measured phenomena via computation from acquired data from the connected physical sensor [12].

We propose to use terminology in fog computing, defined as computational devices controlling and making complex analysis above the other edge devices connected in the same local network. As fog computing, we can use all devices mentioned in the edge computing paradigm as well as personal computers (PC) or server computers. The condition is to design enough computational capacities for complex algorithms, methods and analysis [24].

**Figure 2 sensors-23-04469-f002:**
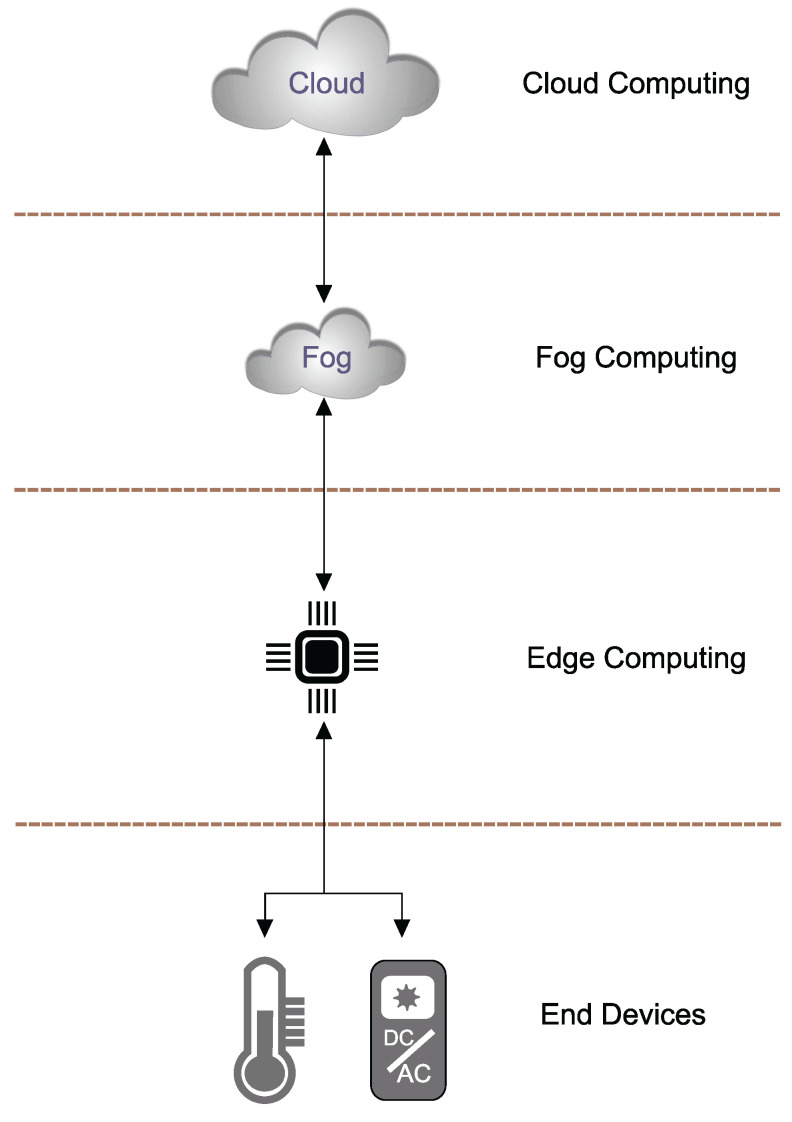
The architecture of IoT computing distribution [25].

### 2.3. Cloud Computing

The cloud computing paradigm has gained popularity in the last decade. In the literature, we can find many definitions of this concept. These can be defined as a computing paradigm serving the central complex computations and storage in specialized data centres. Cloud has a centralized approach, and the fog has decentralized control over the edge devices and sensors. Hereby, the centralized approach in cloud computing from the side of software abstraction. Physically, the applications implemented in the cloud could run across many connected computational devices. In cloud computing, all connected devices have to behave as one computational device. The user should not recognize the distribution of computational capacity across nodes between servers in cloud applications [23,26].

## 3. Transportation Sensoric Systems

Most countries and cities try to optimize the traffic and transportation control via IoT concepts to develop smart cities and smart transportation concepts. A lot of different routes have implemented IoT concepts. Still, the development of new sensors, computational devices, and protocol concepts create new possibilities to improve the effectiveness of these systems including maintenance, financial and computational capacities [27,28]. The mentioned tasks in transportation have their specifications creating a constraint to install some kinds of sensors. One of the vital requirements of the sensor is the ability to produce actual data where the control unit considers these data. In the intelligent transportation system, we use physical sensors categorized into electrical sensors in our taxonomy mentioned in Section 2.1. There could be created either a virtual sensor based on these physical sensors. In this section, there is describe a survey of the most common sensors used in transportation and traffic systems and their taxonomy shown in Table 1 from the view of deployment.

In real-time data analysis, intelligent transportation systems use sensors that measure humidity, temperature, wind speed, precipitation, and visibility. They play a critical role in forecasting weather conditions that impact traffic safety. However, this paper does not include the sensors used for the weather forecast in the taxonomy of sensors mentioned in Table 1 because they do not solve tasks based on motion detection and object tracking.

### 3.1. Intrusive Sensors

Intrusive sensors are sensors installed on pavement surfaces. Intrusive sensors are also categorized as in-roadway-based sensors. They are accurate for vehicle classification performance for transportation monitoring tasks and processing other data types such as speed or weight measurement [29].

#### 3.1.1. Piezoelectric Sensors

Piezoelectric sensors are based on piezoelectric materials such as barium titanate (BaTiO3) or lead titanate (PbTiO3). These materials generate an electrical signal when subjected to mechanical impact or vibration. The measured signal is proportional to the force or vehicle weight [19].

#### 3.1.2. Fiber Bragg Gratting Sensors

The fiber Bragg grating (FBG) sensor is used in civil engineering applications to measure temperature, strains, and loads [30,31]. FBG sensors are sensitive to damage from careless manipulation throughout the installation process because they are built from silica material. There are used many kinds of materials for packaging. The packaging material, made from glass-fiber-reinforced polymer, provides reliable protection to the FBG sensor. The reliable packaging material enables the use of FBG sensors for monitoring the traffic situation in the intelligent transportation system [32].

#### 3.1.3. Pneumatic Road Tube

A pneumatic road tube placed on the road’s pavement being connected to the sensor mechanism. This sensor’s mechanism is proposed for producing an electrical signal from a burst of air pressure created when the vehicle’s tires pass over the tube. The received signals from this sensor can classify and count vehicles passing through the road section [33].

#### 3.1.4. Vibration Sensor

It is possible to build vibration sensors using various technologies. Choosing the optimal materials depends on the environment and the degree of sensitivity that we have to use to implement the vibration sensing system. The study [34] proposed using the road pavement as a source of vibration transduced to the electrical signal. Microelectromechanical system (MEMS) accelerometers transduce generated vibrations to electrical signals. A control unit processes these electrical signals to classify and count the vehicles on the road section.

#### 3.1.5. Loop Detector

An inductive loop detector is a coil of metal wire. The wire captures changes in inductance. Changes in inductance generate a time-variable signal of vehicles passed through this detector. According to the generated signal, the computation unit can classify the vehicles’ presence and categories via amplitude, phase, and frequency spectrum analysis [35].

Depending on the installation method, there are two types of loop detectors: saw-cut and performed loops. In the saw-cut method, the installation process needs to saw-cut the pavement to lay the loop wires under the pavement. The loop wires require protection by filling the pavement after installation. The performed loop detectors are not installed under the pavement. In this method, instead of saw-cut pavement, the loop wires are installed inside the PVC pipe. The PVC pipe is attached to the pavement [36].

#### 3.1.6. Magnetic Sensors

Every vehicle’s construction consists of ferrous metals inducing disturbance to the Earth’s magnetic field. The function of magnetic sensors is to measure the Earth’s magnetic field distortion caused by passing vehicles through the sensor. Magnetic sensors are more energy-efficient with smaller size, weight and cost than inductive loop detectors [37].

### 3.2. Non-Intrusive Sensors

Non-intrusive sensors are not installed in the pavement surface to compare intrusive sensors. They could be installed on the sides of or over the roadway. Non-intrusive sensors are more expensive to produce than intrusive sensors. However, the installation process of non-intrusive sensors is cheaper than that of intrusive ones because saw-cutting or digging under the road surface is considered technically difficult and expensive. The maintenance of road surfaces with installed intrusive sensors as well as standalone intrusive sensors requires an expensive technological approach. In conclusion, the development cost of reliable non-intrusive sensors for intelligent transportation tasks, including the installation, maintenance and service costs, might be cheaper than intrusive sensors. Still, non-intrusive sensors could classify with less accuracy because weather conditions and other environmental events not connected to the transportation could significantly impact classification accuracy processing of other traffic monitoring information such as speed measurement or free parking slot management [29].

#### 3.2.1. Radio Frequency Identification

Radio frequency identification (RFID) sensors can be categorized into two main categories:RFID with passive tags;RFID with active tags.

RFID with passive tags consists of two components. The first component is the RFID reader, which reads (senses) data and sends an electromagnetic signal. The second component is the RFID tag, which transmits the data stored in a memory chip via the specific radio frequency signal. The passive RFID tags do not use an internal power source. They consider obtaining electrical energy from the RFID reader’s electromagnetic signal conducted to electricity. Passive RFID tags can only be applied for short distances due to the fact that they miss power source limits by the reader’s electromagnetic signal strength.

The RFID with active tags works similarly to the RFID with passive tags. The difference is the working principle of the RFID tag. Active RFID tag has an internal power source and automatically transmits the data to the reader after connecting to the electricity. We can use active RFID tags for longer distances than passive RFID tags [38].

RFID sensors are used for free parking slot management or vehicle identification. Active RFID tags enable the transmission of a vehicle’s diagnostics about the expected service date or daily fuel consumption. Vehicles can then be located inside tunnels via active RFID tags. This solution is a good alternative to the GPS navigation system because GPS navigation does not work inside tunnels or buildings [39].

#### 3.2.2. Acoustic Sensor

Acoustic sensors are based on an array of microphones. They capture audio signals induced by passing vehicles. For vehicle classification, we have to extract features from the captured sound. This task is challenging because a sound can contain much noise, and every vehicle has its specific sound. The production of electromobility cars responds to the fact that these cars might be more silent than traditional ones using petrol or diesel vehicles. Acoustic sensors can be used in addition to other sensors [40].

#### 3.2.3. Laser Scanner

A laser scanner is a sensor measuring the distances between its position and the object of interest via a laser beam. Laser scanners are divided into several categories depending on the working principle of distance calculation. One type of laser scanner is a light detection and ranging (Lidar) scanner. The Lidar working principle and its categorization are described as a standalone sensor in our taxonomy (Section 3) because Lidars use similar techniques to radars for distance measuring [41]. However, comparing the radars, Lidars use lasers instead the radio waves. As mentioned above, there exist other types of laser scanners except Lidars. Those types of laser scanners are based on triangulation, time-of-flight (ToF), phase-shift and structured light scanners [42].

Laser scanners based on triangulation use a determined camera position, laser position and angle of laser beam emission to calculate the point’s depth.

ToF laser scanners measure the time of laser beam flight to the object of interest. Regarding the measured time, the ToF laser scanner computes the distance between the laser beam emitter and the object of interest. ToF scanners use a camera or similar photosensitive device to simultaneously measure the depth of many points in the camera’s specific narrow range of visibility.

Phase-shift laser scanners use a continuous laser beam. The sensor inside the scanner calculates the distance via phase-shift of emitted and reflected waveforms of the laser beam.

Structured light laser scanners use a pattern emitted via a laser beam onto an object. The deformation of pattern shape onto an object’s surface is processed through the camera system.

The laser scanner is used to take the shapes of vehicles via a calculated depth map [43]. Laser scanners based on ToF or the phase-shift principle are more expensive than cameras, and laser scanners produce data of lower quality than cameras [41].

#### 3.2.4. Radar

Radio detection and ranging (Radar) is a sensor that measures the presence and distance of various objects. The main principle consists of transmitting and reflecting radio waves from scanned environments [44]. Radar is less accurate than Lidar. Radar provides only a 2D map of object localization, and Lidar provides a 3D representation of the scanned environment. On the other hand, Radar is less weather-sensitive than Lidar [41].

#### 3.2.5. Wi-Fi

A wireless field (Wi-Fi) has a purpose as a network protocol for sharing internet access by radio waves. This protocol was used for collecting data along highways. The principle of using radio waves for data exchange between devices enables the localisation of the connected device through the strength of the signal from the Wi-Fi transceiver [45]. Comparing the signal’s strength between the Wi-Fi receiver and transmitter also enables detection of vehicle motion [46,47].

#### 3.2.6. Lidar

Lidar works on similar principles to radar, but it has some specifications. Some light sources are used, commonly lasers, to transmit the signal into the environment, and photosensitive elements receive the reflected signal. The calculation of the object’s range, presence, and shape is the same as in the Radar sensor mentioned above [48].

#### 3.2.7. Passive Infrared Sensor

A passive infrared sensor (PIR) detects the object’s presence by generated heat energy from vehicles or other objects. Every object that generates heat shines a ray in the infrared spectrum. A Fresnel lens focuses the ray onto the sensing element, which transforms the infrared signal into an electric signal. The PIR sensor is widely used for motion detection in a small range [49].

#### 3.2.8. Ultrasonic Sensor

Ultrasonic sensors use sound waves transmitted at high frequencies not audible to human ears to compute distances between objects. Sensors detect reflected sound wave energy and convert it into an electric signal. The control unit has to process the electric signal. It measures the time difference between transmitted and received sound waves to calculate the distance of the obstacle. Ultrasonic sensors are widely used for measuring vehicle flow or free parking slot management [50,51]. The main disadvantage of ultrasonic sensors are their high sensitivity to environmental noises [52].

#### 3.2.9. Camera Systems

Cameras are image sensors (Figure 3). The sensor captures images in 2D that could reconstruct a 3D object with special techniques, such as stereovision. All image sensors consist of a photodetector array called pixels. These pixels transform electromagnetic waves in the visible light spectrum into electric signals [53]. The methods of transforming visible light into electric signals divide the image sensors into two types: charged coupled device (CCD) and complementary metal oxide semiconductor (CMOS) [54].

CCD sensor pixels consist of individual metal oxide semiconductor (MOS) diodes. These diodes are arranged in rows and columns. The rows and columns create a matrix where the energy of visible light charges every diode proportional to the light’s intensity. The control unit in the CCD camera uses a mechanism to transform the diode’s charge from each pixel in the matrix to the electric signal. The CCD image sensors take pictures with high quality. The disadvantage of CCD cameras is the processing method of pixels. The camera’s control unit has to process every pixel sequentially to reconstruct the image, which is time-consuming and energy-consuming [53,54,56].

CMOS sensor pixels are composed of photodiodes and CMOS transistors. Every CMOS transistor works as a switch generating a digital electric signal from transformed visible light. The main advantage is the possibility of parallel processing of every pixel by the camera’s control unit. It increases the speed of taking pictures with low power consumption. The CMOS image sensors are cheaper than CCD image sensors [57].

The increased quality of camera sensors and suitable computer vision (CV) methods and algorithms enable us to process and classify many tasks in intelligent transportation. The possibility of camera sensors is deeply described in Section 5 and Section 6.

### 3.3. Vehicle Sensors

The reliability of autonomous or semi-autonomous vehicles depends on collecting the actual data from the traffic situation. It is necessary to install sensors in the vehicle to obtain actual data from the road. Table 2 shows the taxonomy of in-vehicle sensors and non-vehicle sensors. This paper defines in-vehicle sensors as sensors that could be used inside a vehicle’s construction to monitor the vehicle status or environment around the vehicle. Non-vehicle sensors are defined as sensors that cannot be installed into a vehicle’s construction, and are only used as part of road infrastructure.

In-vehicle sensors can also be used to transform cars into mobile sensor systems in the IoT concept of communication vehicle to infrastructure (V2I). The possibility of using a car as a sensor is described in (Section 5).

## 4. Transportation Tasks Based on Motion Detection and Object Tracking

This section describes the main transportation tasks used for monitoring traffic or infrastructure. The monitoring tasks are focused on the object’s motion detection and object tracking. Automatized monitoring tasks via IoT concepts in transportation could improve the conditions of traffic situations.

In-vehicle sensors could detect or warn the driver before a collision with other objects [58]. Development and research in artificial intelligence (AI) applications enable the building of driverless vehicles that consider data collected from sensors in real-time and process these to control vehicles while maintaining security standards [59].

### 4.1. High-Occupancy Vehicle Lane Management

Several countries try to reduce traffic congestion by motivating citizens to share cars or use public transportation. For this purpose, high-occupancy vehicle (HOV) lanes have been built, which serve only vehicles with two or more passengers. These HOV lanes enable drivers to reach their target destination quickly, avoiding potential traffic congestion [60]. HOV lane management systems are proposed to check the count of passengers in every vehicle. The system sorts the vehicles that have permission to use the HOV lanes. There are two options for checking the passenger count in the vehicles. The first one is using sensors for person detection from the checking station [61]. The second one is using seat occupancy sensors installed on the car’s interior [62]. The information extracted from the sensors could be sent to the checking station via the dedicated communication protocol.

### 4.2. Incident Detection

We detect incidents via various methods and different systems. Transportation incidents could happen in various scenarios, such as a vehicle fire inside a tunnel [63], traffic jams resulting from a damaged car [64], or chain car accidents [65]. For this purpose, many different types of sensors and communication devices are used to contact emergency services. We can divide incident identification into two aspects. The identification of incidents via infrastructure—for example, the camera surveillance system in a tunnel identifies a fire in a tunnel, which alarms the operator to begin the rescue of people inside the tunnel [66]. The second possibility is using in-vehicle sensors. For example, the ultrasonic sensors detect the collision, and the car’s board computer contacts the emergency service by wireless internet connection [67].

### 4.3. Vehicle Counting

Many paid highways use a counting vehicle system to monitor usage [68]. The counting vehicle system produces information about the frequency of visiting the routes. This information is used in future planning for maintenance or building new highways or roads [69,70]. The advanced counting vehicle system categorizes vehicles into many categories, such as buses, trucks, or cars (Section 4.5). These collected data could help manage traffic.

### 4.4. License Plate Recognition

License plate recognition is an essential task in road surveillance. This task can identify cars with detected anomalous behavior. We can usually apply the methods of automatic license plate reading after finding the car’s position via some object detector. According to [71], the automatic license plate recognition (ALPR), process is divided into three steps:Localize the position of the license plate in car.Segment characters from the background.Apply optical character recognition (OCR) methods.

This task is essential for checking the priced parking places [72] or identification of stolen cars [73].

### 4.5. Vehicle Type Classification

Many roads have some restrictions on the types of vehicles they allow because of safety, flow, and maintenance costs. Trucks, due to their heavy weight, are prohibited from using some routes [74]. In many countries, truck drivers have to pay extra charges to use highways due to higher maintenance costs [75]. Sometimes, the large size of trucks or buses limits passing through some sections without a traffic incident [76]. The data collected from the categorized count of vehicles by particular sections of the roads can improve the management of traffic lights to avoid traffic jams [66,77]. This information can be used by the dispatchers of public transport to react to special events.

### 4.6. Speed Measurement

Safety is considered a target priority in transportation. High-speed driving vehicles increase the risk of an accident. The increased kinetic energy of high-speed driving impacts the crash consequences after the collision with another vehicle or object. The risk of accidents in high-speed driving is related to limited human reaction time. States worldwide use speed limit restrictions in traffic management to reduce accident risks [78]. One of the aims of an intelligent transportation system is to check adherence to speed limit restrictions. The police use sensor systems that detect vehicle motion and track vector movements for speed measurement [79].

### 4.7. Parking Management

The increased number of people buying the cars influence the fact that they are widely used as a kind of individual transport method. Many people ride to work, school, or shop by car, occupying parking slots across a city or town. Free parking slots are a limited resource in many cities; therefore, many parking houses or places are priced [80]. Priced parking places require some automatic system of checking all the car owners to pay for parking slots [72]. A lot of automatic parking place checking is based on a sensor system similar to the vehicle counting system.

### 4.8. Weight Measurement

Worldwide, there are many specialized sections of roads that have specific vehicle weight constraints. These constraints exist because of road maintenance or security reasons [74]. To avoid the entry of overweight vehicles, sensors are installed that measure the vehicle’s weight, and the system can close the entry gate. For example, bridges are places where a vehicle’s axle weight restriction is necessary to keep traffic safe [81].

## 5. Applications the Sensor System in Transportation Tasks

In this section, we introduce the transportation monitoring tasks categories mentioned in (Section 4), which could be solvable sensor systems, as mentioned in (Section 3). Table 3 represents state of the art in sensor systems used for intelligent monitoring transportation tasks in the last decade. The literature review in Table 3 refers to identification of research challenge for each sensor system in tasks mentioned in the table. Table 3 uses a literature survey and review to compare the transportation tasks mentioned in (Section 4) and the sensor system mentioned in (Section 3).

Table 3 describes the potential of mentioned sensors to solve transportation monitoring tasks based on motion detection and object tracking. From the sensors mentioned (Section 3), we can conclude that we may solve many monitoring transportation tasks via the camera system in almost all applications except for weight measurement. CV methods and algorithms enable the creation of a virtual sensor (Figure 1) from the camera to sense many objects. The research’s progress in traditional CV methods and artificial deep neural networks makes motion-detection and object-tracking methods more effective [157,158].

Intrusive sensors such as magnetic sensors, loop detectors or pneumatic road tubes use a specific installation configuration for each purpose (Table 3). This installation configuration impacts the detector sensitivity level. Each monitoring task requires a different sensitivity level. For example, single- and dual-loop detector configurations have different sensitivity levels. The parking slot management system does not need to categorize the vehicle; therefore, the single-loop detector is efficient for parking slot management. In monitoring highways, the information about counted trucks and other vehicle types has a significant role. Dual-loop detectors increase the vehicle categorization accuracy [41].

FBG sensors are the most prospective intrusive sensors (Table 3). Suitable installation configurations and machine learning models can classify many events. The disadvantage is a calibration system. The advantage of the FBG sensor is the low-cost manufacturing investments compared with metallic sensors, long service life, and low maintenance cost. The FBG sensor is categorised as a virtual sensor source for traffic monitoring tasks in intelligent transportation systems because it measures a reflected light spectrum due to the fibre strain changes. It requires a complex, trained machine learning model to filter and extract valuable data for vehicle presence, speed estimation or weight measurement [32,159].

This article proposes categorising intrusive sensors, except FBG sensors, into physical sensors because the installation configuration classifies the vehicle presence directly via simple signal processing. These sensors consider common vehicle features that every vehicle consists of metal parts. The metal parts change the magnetic field’s behaviour (magnetometer, loop detector) or impact the vehicle’s weight characteristic (pneumatic road tube) [41].

In conclusion, intrusive sensors except FBG sensors have limited usage for collecting complex data about the actual situation on the road.

Article [2] categorises a camera system, Lidars, Radars, and passive infrared sensors as vision-based sensors. These sensors produce an output in map or image form. We propose categorising these sensors as virtual sensors for intelligent transportation system monitoring tasks because CV methods can extract valuable information from vision-based sensor data (Section 6). Sensors such as cameras or Lidars require other types of CV methods than radars or PIR sensors. PIR sensors detect a level of infrared radiation. Thresholding or neural networks are the only CV methods that can effectively use a control unit from data obtained from one PIR sensor. PIR sensors use thresholding for motion detection, but the sensors do not recognize what makes a motion [160]. Radar produces a 2D map of an object’s shapes and positions. On this map, we can use pattern recognition and advanced feature extraction approaches from CV methods, such as corner points or edge detectors [161,162]. Lidar can make a 3D map of the sensed environment. The measured points of the Lidar’s 3D map with the suitable transformations enable the reconstruction image with shades of selected colours palette where the shades represent the measured points’ distance from the selected perspective. The reconstructed image can use similar CV methods to images made using camera systems [163]. However, the Lidar scanning depth and missing objects’ colour data do not enable using all CV methods.

In Table 2 (Section 3.3), we mention the division of sensors by non-vehicle and in-vehicle sensors. and we used in-vehicle sensors in vehicle-to-vehicle (V2V) interaction or V2I interaction. Non-vehicle sensors are used for infrastructure monitoring systems [1]. In-vehicle sensors are served for the driver’s safety management to avoid collision with another object [164]. The Wi-Fi system can serve not only as a sensor system but also as a communication way to interact with emergency services in case of an accident [1]. Recently, many companies have tried to implement autonomous driverless vehicles that have to gain complex information about the actual situation from the road environment. Sensors such as ultrasonic or radar sensors are inefficient in obtaining the necessary data for autonomous vehicles [2]. Ultrasonic sensors have a small range of detection objects, and Radar has less accuracy than Lidar. For Lidar, it is possible to reconstruct the 3D scene for long distances, which can help to avoid collisions with other vehicles on the road. The reconstruction of the 3D scene is also possible with camera systems, achieving similar results to Lidars [42]. Furthermore, Lidars cannot read a traffic sign, where a camera system can.

Referring to Table 3, intrusive sensors such as loop detectors or magnetometers are limited to use in the HOV lanes management system. However, almost all vision-based sensors can check a vehicle’s occupancy status. Despite the fact that vision-based sensors are suitable for this purpose, there are still some problems. Lidars and laser scanners are too expensive for installing to the car’s interior only for counting the passengers. An automatic passenger-counting system based on a camera system needs to be installed at distances from which the person’s face could be recognizable. This approach is intrusive. The optimal solution is using sensors detecting passengers via seat occupation and sending the passenger count from the vehicle with V2I protocol to the HOV lane station.

The parking assistance system is one of the standard systems being implemented in cars. This system helps the driver search for a free parking slot and successfully park without collision. Parking houses can monitor free parking slots with a huge number of intrusive sensors that are costly to implement and maintain or with few cameras which can cover the same area with cheaper investment. Cameras can navigate cars across a parking lot and help calculate the trajectory necessary to reach a free parking slot. Parking management systems will become a vital part of the intelligent transportation system, with which V2I will cooperate [80,165,166,167].

Intrusive sensors have limitations in gaining complex data about the actual situation in monitoring road tasks. All intrusive sensors have a smaller distance detection range than non-intrusive sensors. These sensors can reliably categorize and count vehicles, and some of them, such as a magnetometer or FBG sensor, can measure speed. On the other hand, the expensive installation process limits their multiple usages in the same route. The most significant advantage of intrusive sensors is that they can only measure a vehicle’s weight. However, not all intrusive sensors can measure a vehicle’s weight, for example, loop detectors or magnetometers.

FBG sensors can monitor almost all tasks mentioned in (Table 3) except the ALPR task. This sensor can be used as a universal intrusive virtual hardware-based sensor for intelligent transportation systems tasks because it can solve most tasks from intrusive sensors (Table 3).

Camera systems visually observe roads by making video records. CV methods can extract a lot more information from these video records. High-resolution images or videos increase details and data quality for the following analysis process. Optimized convolutional neural network (CNN) models detect many objects of interest via classification rectangle (object detection methods) [168]. Object-detection methods are used for almost all monitoring tasks (Table 3). The CV methods that run on the camera system videos form a universal non-intrusive virtual sensor in the intelligent transportation system (Table 3).

## 6. Universal Sensor

This literature survey in Section 5 concludes and summarises the possibility and flexibility of the sensors mentioned in Section 3. Many camera systems connected to computational resources in edge, fog or cloud servers provide the specific purpose of monitoring tasks in the intelligent transportation system. The camera system produces image data containing useful information for selected monitoring tasks in Section 4. On the same image or video stream, other CV methods and algorithms can be simultaneously applied that extract or sense different events or objects independently. Table 3 summarises the camera system’s usability as well as the usability of other sensors for selected monitoring tasks (Section 4). Table 3 concludes that the camera system is a non-intrusive sensor that can solve most tasks by comparing other sensors. Camera system can be a declared as universal sensor for monitoring tasks because mentioned reasons above (Section 4).

The construction sensor system consists of a suitable surveillance camera and embedded SBC to the camera with Linux distribution. The embedded SBC in the camera allows the creation of a universal software-based virtual edge-enabled intelligent sensor system (EEISS) for most monitoring tasks (Section 4) in the intelligent transportation system. The purpose of the EEISS depends on the implemented and uploaded software solution based on plug-in (service-oriented) architecture. Each monitoring task (Section 4) has its own software plug-in (service) for processing and classifying the same camera’s data. The plug-ins (services) provide the flexibility of purpose EEISS based on the requirements for the system. In our case, the plug-ins (services) represent the software-based virtual part of the sensing system in the EEISS. For example, independent plug-ins (services) can be the speed estimation, ALPR or the vehicle’s motion detection and tracking software embedded into EEISS. The EEISS also allows additional data from other sensors, such as FBG sensors or Lidar, to these software-based plug-ins (services). In Section 7, we introduce the case study scenario of the development and usage of the EEISS.

## 7. Case Study Scenario

In this section, we present a case study that shows the effectiveness of the EEISS in providing complex information from selected monitoring tasks in the intelligent transportation system. The EEISS allows significantly improved flexibility of the sensor system for mentioned monitoring tasks. This flexibility helps to fill the market gap where a similar solution to the EEISS in the market is not available at this moment. This case study aims to prove the achievability of developing and building the EEISS for intelligent transportation system purposes. The case study consists of the proposed architecture (Figure 4) based on a state-of-the-art literature review, implementing the architecture’s prototype and demonstrating the first experimental results in real-life conditions.

We implemented and deployed the testing of a mast-mounted surveillance station based on this architecture in real-world conditions on a selected highway. Our research group can upgrade and optimize hardware and software solutions for this testing station, including changing various vision-based sensors according to their effectiveness. The testing station architecture is described below in 5 points.

(1) The intrusive pressure sensor (for example FBG sensor) is installed on the road surface to measure the vehicle’s weight. The sensor sends data to the fog computing layer (4) for following analysis.

(2) The mast-mounted surveillance system consists of a camera system. This system sends a video stream to the edge computing layer (3).

(3) The edge computing layer consists of SBCs embedded in the camera system construction (2). SBCs run firmware with computer vision methods and algorithms directly into the Camera System. These methods and algorithms of vehicles detect motion and classify the situation on the road. If the SBC system detects a vehicle presence, it starts the ALPR system or other services to check the traffic situation mentioned in (Section 4). The edge computing layer sends the classified, collected and aggregated data to the fog computing layer (4) to analyse the complex situation.

(4) The fog computing layer represents the microserver connected near the mast-mounted surveillance system. The microserver directly connects the intrusive pressure sensor and the system of SBCs with its cameras (EEISS). This microserver collects sensor data and controls the data flow to store or send to the cloud computing layer.

(5) The cloud computing layer uses a rent server to provide many applications for traffic monitoring and management system. It stores data for the following data mining projects or contacts emergency services in case of accidents.

The proposed and built architecture demonstrated the interesting first partial results described below in Figure 5. The picture on the left in Figure 5 shows the yellow rectangle (1) representing the road surface’s free slot from vehicle presence. The picture in the middle of Figure 5 describes the same rectangle (1) with green colour representing a positive motion detection of vehicle presence. After positive motion detection, the SBC’s program takes data from the intrusive piezoelectric sensor (2) in Figure 5 about the vehicle’s weight or category and starts the ALPR program from another camera. The ALPR camera’s picture is shown on the right of Figure 5.

The piezoelectric sensor has been installed into the road surface in this route for over two years, so the firmware is calibrated with high reliability to measure the weight and classify the vehicle type. The firmware accuracy for both tasks is over 97%. There is tendency to use the accuracy of piezoelectric sensors for the automatic annotation of image-based data from the cameras for machine learning purposes. Collectin more annotated data increases the machine-learning model accuracy. On the other hand, the piezoelectric sensor cannot be used for reliable speed measurement and other tasks mentioned in (Section 4). In the following research, we plan to replace the piezoelectric sensor with the FBG sensor to solve more monitoring tasks in intelligent transportation systems.

The installed camera systems showed us as a reliable monitoring system for many tasks mentioned in Section 4. In future work, we plan to buy a high-speed camera system to enable us to collect data to estimate vehicles’ speed. We also plan to try different angles of installation as well as to buy cameras with night vision mode. Cameras with night vision mode can reduce the impact of extreme weather conditions at night to obtain image data usable for CV methods.

## 8. Discussion

The research community has made significant progress in developing various sensor systems for intelligent transportation. A huge number of mentioned physical sensors directly measure the observed values and classify the observed events through these values. Sensor systems, except the vision-based sensor systems, mostly focus on a small range of transportation tasks.

Camera systems can scale sensitivity via various lenses, sampling rates, and resolutions. These input conditions enable us to create an universal virtual sensor for many tasks with a wide range of data quality input. In last decade, scientists developed many open-source as well as paid frameworks for pre-trained neural network models for image data. We can also use non-machine learning approaches in CV methods to solve tasks requiring motion detection and object tracking.

The semiconductor industry enables us to manufacture strong SBC and MCU edge devices where we can deploy the neural network models as well as other CV approaches. The edge devices are also small in size, with small energy consumption. These edge devices can be connected directly to the same places where the Camera Systems are installed. The connection of edge devices (with deployed CV methods or algorithm software) that are used as a virtual sensor, and a Camera System, which is used as a physical sensor, create a new physical sensor to sense or measure new phenomena such as speed estimation, fire detection, or distance measuring. However, edge devices have limited computational power to compute complex machine-learning models in real-time applications.

Pre-trained neural network models are unreliable for images taken in extreme weather conditions. To improve theses models’ accuracy in extreme weather conditions, the models need to learn from the additional dataset. The dataset can be collected using a vehicle’s cameras. The car’s camera can collect a dataset from traffic situations in extreme weather conditions from various view angles that can be used to improve existing machine learning models.

To obtain this dataset, engineers have to implement communication interfaces and protocols in the infrastructure’s monitoring system. The communication protocols are to exchange valuable data between V2I and V2V that need to be standardized, and they are in development. A reliable, standardized protocol can be used in every vehicle as the sensor data source for monitoring traffic in intelligent transportation monitoring tasks.

Creating a universal intelligent sensor system for monitoring tasks in the intelligent transportation system is one of the research challenges. In the following research, we focus on a camera-based system. To this purpose, it is necessary to find a suitable camera system with parameters such as resolution, sampling rate, and other features that will help to achieve this aim. Then, we should propose, develop and test a communication protocol between SBCs or MCUs. Based on these conditions, the camera system classifies the monitored tasks mentioned in Section 4. If there is a need to use complex machine learning models in real-time applications installed in SBCs or MCUs, then the computational limitations of SBCs or MCUs have to be solved with hardware-based or software-based techniques to simplify the model computation. Hardware-based inference acceleration techniques increase the power consumption and make the system less compact. Software-based techniques focus on the simplicity of machine-learning models or model explainability techniques to extract features programmatically. The appropriate SBC or MCU selection can reduce the machine-learning model or other algorithm optimization via hardware-based or software-based techniques. Improvement of the existing pre-trained machine-learning models for extreme weather conditions requires installing the camera-based monitoring system in many places around the world to obtain data or obtaining data from vehicles’ camera system records. Standardizing the V2I communication protocol is necessary to get the vehicle’s sensor data and camera records regarding the driver’s security and privacy. Integrating all necessary data including the extreme weather conditions to make an efficient machine-learning model is a technically complicated process, as well as the training model process of the suitable machine-learning architecture model. SBC or MCU have to use the optimized variant of the trained model. We propose using the traditional CV methods, not using the machine-learning approach and not requiring a complex dataset to make the CV’s software solution [169,170].

## 9. Conclusions

Sensors are the essential element of IoT in collecting data from the physical world to the digital form. Implementing IoT concepts into transportation can improve monitoring systems for the maintenance of roads. The quality of the road’s pavement impacts traffic safety and fluency. The improved Lidar and camera manufacturing technology partially solved the research challenges in monitoring roads. There are many real-time algorithms and methods to recognise vehicles and pedestrians from Lidars and camera data. These algorithms and methods are suitable for use in edge devices. The edge device’s optimised firmware based on CV methods can be used as a universal physical sensor based on the fusion software-based virtual sensor and camera or Lidar as a physical sensor. Cameras have the disadvantage of taking photos in bad weather conditions. Lidars could solve this disadvantage and use similar CV algorithms and methods to classify the motion detection of vehicles or pedestrians and track their movement.

In this study, we performed a literature survey of the available sensor systems used for intelligent transportation system purposes. These sensor systems were categorized according to our proposed taxonomy from many angles of view. We also surveyed the monitoring tasks in the intelligent transportation system based on motion detection and object tracking methods. The taxonomy and literature survey of the sensor systems, as well as monitoring tasks, were used for making Table 3, which reviews the sensor systems’ effectiveness for each monitoring task.

The result of this review was the case study where we proposed the architecture of EEISS. The EEISS uses vision-based sensors to create a flexible universal sensor system for monitoring tasks in the intelligent transportation system. The flexibility of the EEISS provides software-based plug-ins (service-oriented) architecture. These software plug-ins (services) use CV methods and algorithms to classify and process image data. These plug-ins (services) are the kind of software-based virtual sensors for monitoring tasks such as ALPR, vehicle counting and classification or speed measurement.

The following research steps focus on optimizing CV methods and algorithms for real-time applications capable of the SBCs’ performance. Our aim was to discover minimal SBC hardware requirements for some selected software-based plug-ins (services). It will be compared the performance of traditional CV and deep neural network approaches in the vehicle’s motion detection and object tracking tasks suitable for SBC from inference time and reliability. To gain reliable image data from the mast-mounted surveillance system, various camera systems will be tested with different frame rates, lenses, angles of the view and resolution matrix. The analytics have used intrusive sensors to count and classify types of vehicles for the last two decades because they are reliable. Furthermore, installing and maintaining intrusive sensors on the road surface is expensive. Therefore, we aim to develop and implement plug-ins (services) based on CV methods in which the classification and data process accuracy will be similar to or better than intrusive sensors. The EEISS based on these plug-ins (services) could replace some intrusive because maintenance and installation costs could be cheaper of intrusive sensors.

## Figures and Tables

**Figure 1 sensors-23-04469-f001:**
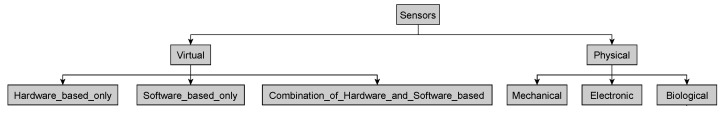
The taxonomy of sensors.

**Figure 3 sensors-23-04469-f003:**
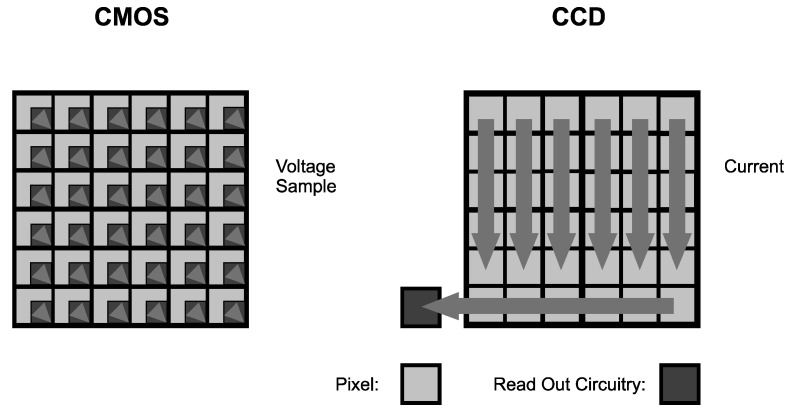
The comparison of CCD and CMOS sensors [55].

**Figure 4 sensors-23-04469-f004:**
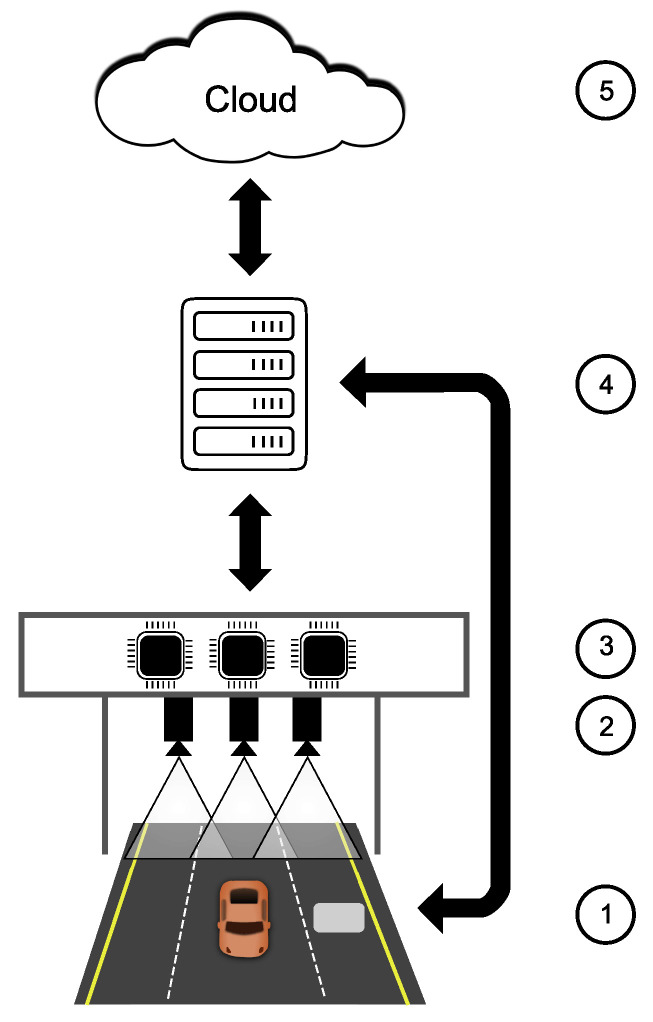
The architecture of proposed intelligent road’s infrastructure monitoring system based on the mast-mounted surveillance station and intrusive sensors.

**Figure 5 sensors-23-04469-f005:**
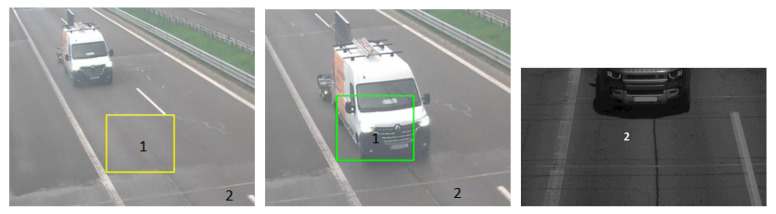
1—The area of motion detection interest, 2—Saw-cut area of intrusive sensor presence.

**Table 1 sensors-23-04469-t001:** Taxonomy of the sensors used in transportation.

Intrusive Sensors	In Roadway Based Sensors	Piezoelectric Sensor
		Fiber Bragg Gratting Sensor
		Pneumatic Tube Detector
		Vibration Sensor
		Loop Detector
		Magnetic Sensor
Non-intrusive Sensors Sensors	Side Roadway Based Sensors	RFID
		Acoustic Sensor
		Laser Scanner
		Radar
		Wi-Fi
	Over Roadway Based Sensors	Lidar
		Passive Infrared Sensor
		Ultrasonic Sensor
		Camera Systems

**Table 2 sensors-23-04469-t002:** Taxonomy of the sensors that can be used in vehicles.

Non-Vehicle Sensors	Piezoelectric Sensor
	FBG Sensor
	Pneumatic Tube Detector
	Loop Detector
	Magnetic Sensor
	Acoustic Sensor
	Laser Scanner
In-Vehicle Sensors	Vibration Sensor
	RFID
	Radar
	Wi-Fi
	Lidar
	Passive Infrared Sensor
	Ultrasonic Sensor
	Camera Systems

**Table 3 sensors-23-04469-t003:** Table of the transportation tasks solved via the sensors.

	HOV Lane Management	Incident Detection	Vehicle Counting	License Plate Recognition	Vehicle Type Classification	Speed Measurement	Parking Management	Weight Measuring
**Piezoelectric Sensor**	In-Vehicle monitoring the seat occupancy approach. [62]	Indirectly with the combination of vehicle counting and machine learning approach as well as the sensor installing as a part of the vehicles detect crashes. [82,83,84]	This approach uses simple statistical models for analysis electrical signal generated from the mechanical pressure. [85]	Not found publication articles on using this sensor as a solution to this problem at this moment.	It uses an obtained weight features to categorize the vehicles. [85]	Indirectly, installing this system as a vehicle counter in many places in the same road section. [86]	Parking management system solved via vehicle counting entry in park houses. [87]	It uses the feature of increasing weight impacting on strength of generated electrical signals. [88]
**FBG sensor**	In-Vehicle monitoring the seat occupancy approach. [89,90,91]	Indirectly with the combination of vehicle counting and machine learning approach. [82,83,92]	This approach uses simple statistical models for analysis electrical signal generated from the mechanical pressure. [93]	Not found publication articles on using this sensor as a solution to this problem at this moment.	It uses an obtained weight features to categorize the vehicles. [94]	Signal wave analysis [95].	Parking management system solved via vehicle counting entry in park houses. [96]	It uses the feature of increasing weight impacting on strength of generated electrical signals. [97]
**Pneumatic Road** **Tube**	Indirectly via categorization of vehicles proposed for public transport. [98]	Indirectly with the combination of vehicle counting and machine learning approach. [82,83]	This approach uses simple statistical models for analysis electrical signal generated from the mechanical pressure. [98]	Not found publication articles on using this sensor as a solution to this problem at this moment.	It uses an obtained weight features to categorize the vehicles. [98]	Indirectly, installing this system as a vehicle counter in many places in the same road section. [86]	Parking management system solved via vehicle counting entry in park houses. [98]	It uses the feature of increasing weight impacting on strength of generated electrical signals. [99]
**Vibration sensor**	In-Vehicle monitoring the seat occupancy approach. [100]	Indirectly with the combination of vehicle counting and machine learning approach. [82,83,101]	This approach uses simple statistical models for analysis electrical signal generated from the vibration. [102]	Not found publication articles on using this sensor as a solution to this problem at this moment.	It uses an obtained weight features to categorize the vehicles. [103]	For railway transportation or heavy vehicles, signal processing and analysis are used. [104]	Parking management system solved via vehicle counting entry in park houses. [105]	It uses the feature of increasing weight impacting on strength of generated electrical signals. [106]
**Loop Detector**	Indirectly via categorization of vehicles proposed for public transport. [107]	Indirectly with the combination of vehicle counting and machine learning approach. [82,83]	This approach uses simple statistical models for analysis electrical signal generated from the magnetic field change. [108]	Not found publication articles on using this sensor as a solution to this problem at this moment.	It uses an obtained weight features to categorize the vehicles. [107]	Indirectly, installing this system as a vehicle counter in many places in the same road section. [86]	Parking management system solved via vehicle counting entry in park houses. [109]	Not found publication article on using this sensor as a solution to this problem at this moment.
**Magnetic Sensor**	Indirectly via categorization of vehicles proposed for public transport. [110]	Indirectly with the combination of vehicle counting and machine learning approach. [82,83]	This approach uses simple statistical models for analysis electrical signal generated from the magnetic field change. [111]	Not found publication articles on using this sensor as a solution to this problem at this moment.	It uses an obtained weight features to categorize the vehicles. [110]	Indirectly, installing this system as a vehicle counter in many places in the same road section. [86]	Parking management system solved via vehicle counting entry in park houses. [112]	Not found publication articles on using this sensor as a solution to this problem at this moment.
**Radio frequency** **identification**	Identification of every passenger approach. [113,114]	Indirectly with the combination of vehicle counting and machine learning approach. [82,83]	This approach reads the data from device [115]	Not found publication articles on using this sensor as a solution to this problem at this moment.	All vehicles identified via this system have to carry information about the type of vehicle implicitly. [116]	Indirectly, installing this system as a vehicle counter in many places in the same road section. [86]	Parking management system solved via vehicle counting entry in park houses. [117]	Not found publication articles on using this sensor as a solution to this problem at this moment.
**Acoustic Sensor**	Vehicle’s engine sound analysis correlated to vehicle occupancy. [118]	The typical sound of a vehicle collision detects the incident. [119]	This approach uses simple statistical models for analysis electrical signal generated from the sound. [120]	Not found publication articles on using this sensor as a solution to this problem at this moment.	It uses the engine’s sound to categorize the vehicles. [121]	Signal processing and analysis through neural network. [122]	Parking management system solved via vehicle counting entry in park houses. [123]	It uses the feature of increasing weight impacting on strength of generated electrical signals. [124]
**Laser Scanner**	In-Vehicle monitoring the seat occupancy approach. [125,126]	Indirectly with the combination of vehicle counting and machine learning approach. [82,83]	This approach scans the environment to find vehicles. [127]	Not found publication articles on using this sensor as a solution to this problem at this moment.	It uses spatial parameters information. [128]	It uses the distance change in the time interval to compute the average speed. [129]	The possibility of exact occupancy-free parking slots checking. [130]	Not found publication articles on using this sensor as a solution to this problem at this moment.
**Radar**	In-Vehicle monitoring the seat occupancy approach. [131]	Indirectly with the combination of vehicle counting and machine learning approach. [82,83]	This approach scans the environment to find vehicles. [132]	Not found publication articles on using this sensor as a solution to this problem at this moment.	It uses spatial parameters information. [133]	It uses the distance change in the time interval to compute the average speed. [79]	The possibility of exact occupancy-free parking slots checking. [134]	Not found publication articles on using this sensor as a solution to this problem at this moment.
**Wi-Fi**	In-Vehicle monitoring the seat occupancy approach. [135]	Indirectly with the combination of vehicle counting and machine learning approach. [82,83]	This approach uses the signal interuption for detect a vehicle presence. [47]	Not found publication articles on using this sensor as a solution to this problem at this moment.	It uses spatial parameters information. [47]	Indirectly, installing this system as a vehicle counter in many places in the same road section. [86]	The possibility of exact occupancy-free parking slots checking. [136]	Not found publication articles on using this sensor as a solution to this problem at this moment.
**Lidar**	In-Vehicle monitoring the seat occupancy approach. [137]	The installed sensor could detect the distance that leads to a collision with other object in the traffic. [138]	This approach scans the environment to find vehicles. [139]	Not found publication articles on using this sensor as a solution to this problem at this moment.	It uses spatial parameters information. [140]	It uses the distance change in the time interval to compute the average speed. [141]	The possibility of exact occupancy-free parking slots checking. [142]	Not found publication articles on using this sensor as a solution to this problem at this moment.
**Passive** **Infrared Sensor**	In-Vehicle monitoring the seat occupancy approach. [143]	Passive Infrared Sensor could detect a fire. [144]	This approach scans the environment to find vehicles. [145]	Not found publication articles on using this sensor as a solution to this problem at this moment.	It uses spatial parameters information. [146]	Indirectly, installing this system as a vehicle counter in many places in the same road section. [86]	Parking management system solved via vehicle counting entry in park houses. [147]	Not found publication articles on using this sensor as a solution to this problem at this moment.
**Ultrasonic Sensor**	In-Vehicle monitoring the seat occupancy approach. [148]	The installed sensor could detect the distance that leads to a collision with other objects in the traffic. [101]	This approach scans the environment to find vehicles. [50]	Not found publication articles on using this sensor as a solution to this problem at this moment.	It uses spatial parameters information. [149]	Indirectly, installing this system as a vehicle counter in many places in the same road section. [86]	The possibility of exact occupancy-free parking slots checking. [51]	Not found publication articles on using this sensor as a solution to this problem at this moment.
**Camera Systems**	Persons counting via computer vision approach on dedicated stations. [61]	Object detection, Object tracking, and image classification approach [150,151].	Object detection and image classification approach [152].	Object detection and image classification aproaches. [153]	It uses spatial parameters information. Image classification and Object Detection [154,155].	Object detection and Object tracking [156].	Object detection and other CV methods [80].	Not found publication articles on using this sensor as a solution to this problem at this moment.

## Data Availability

The mast-mounted surveillance system of the Betamont company.

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
