# Peer review of "Review of IoT Sensor Systems Used for Monitoring the Road Infrastructure"

_sensors, 2023, doi:10.3390/s23094469_

Round 1

Reviewer 1 Report

The reviewed manuscript is a review paper that provides a taxonomy of sensors intended for intelligent transport systems (ITS). It offers a general overview of monitoring tasks in the road transport domain, with a specific focus on sensors designed for motion detection and object tracking. The categorization used allows for a comparison of the effectiveness of the mentioned kinds of sensors. The literature survey resulted in the design of a custom architecture, which was created and tested within the framework of a case study. Unlike other similar reviews, this paper considers physical and virtual sensors and data processing at different computing levels (cloud, foggy, edge). One of the strengths of this review is the design of a universal software-based virtual edge-enabled intelligent sensor system (EEISS) architecture. However, the text is burdened with errors and grammatical inaccuracies.

The manuscript is clear, relevant for the Sensors journal field and presented well-structured – beginning with a general survey and taxonomy, comparison and concluding results of own research.

Line 294: The statement “Ultrasonic sensors use sound waves transmitted at frequencies 25 and 50 kHz …” is too categorical and principally not true since there are ultrasonic sensors that operate in the range of 20 to 25 kHz as well. These sensors are typically used for applications that require longer detection ranges or the detection of larger objects (for example Murata MA40S4S - an ultrasonic sensor that operates at a frequency of 20 kHz and has a detection range of up to 5 meters; commonly used in applications such as parking assist systems, object detection, etc.)

The overview of sensors needed for the domain of ITS is not complete. For example, weather stations can also play a critical role by providing real-time data on weather conditions that can impact road safety and traffic flow. They are typically deployed along roadways or in strategic locations to monitor weather parameters such as temperature, humidity, wind speed, precipitation, and visibility. I admit that is not in the category of motion detection and object tracking, discussed in principle in the paper, but it could be mentioned on the side.

Another missing sensors, which could belong to tracking and detection tasks and which should be mentioned and categorized are “high occupancy sensors” - they can help to manage and optimize the use of such called “High Occupancy Vehicle” (HOV) lanes, which are reserved for vehicles with multiple occupants (typically two or more). Again, they use a variety of technologies to detect the number of occupants in a vehicle. Common technologies include infrared sensors, microwave sensors, and video-based systems. The data collected by high occupancy sensors can be used to enforce HOV lane restrictions, monitor compliance with HOV lane regulations, and provide real-time information to drivers via variable message signs. Camera-based HOV approach can be easily included within a concept of the EEISS proposed by the authors.

Cited references come from the period 1954 – 2023. They are relevant, most of them are recent publications (3-7 years old). There is only 1 self-citation – [75]. All references are cited in the text.

Recommendation: include other latest review-like sources related to intelligent transportation systems, their sensors and usage of IoT, especially: 

·       https://www.mdpi.com/1424-8220/23/8/3880 (2023)

·       https://www.ijrti.org/papers/IJRTI2109012.pdf (2022)

·       https://ijettjournal.org/Volume-70/Issue-6/IJETT-V70I6P217.pdf (2022)

·       https://ieeexplore.ieee.org/document/9792638 (2022)

The manuscript is scientifically sound. The case study with the proposed architecture of

EEISS is appropriate to test the concept of a universal ITS monitoring solution.

Comments on the interpretation of Table 2: At first glance (from a terminological point of view), the classification of sensors used in a vehicle into in-vehicle and non-vehicle might be a bit confusing. An additional explanation/definition would be appropriate. If I understand the authors properly, in-vehicle sensors are built-in vehicle sensors while non-vehicle sensors are those additionally installed in a vehicle, e.g. by a user.

 Recommendation: The third photo in Fig. 5 (farthest to the right) could be purposefully edited to increase its brightness and readability (unlike the photos given on the left, its resolution, so necessary for optical character recognition, is sufficient).

I agree with the presented conclusions, they seem to be consistent with the arguments presented within the literature survey.

No ethical issues were identified in the document that needed to be addressed (especially in relation to Figure 5).

English style needs revision. To illustrate it a few samples of typical inaccuracies contained in the text:

2: The internet of things (IoT) ... should be with capitals (The Internet of Things (IoT))

31: Organizing fluently and safe traffic ... an adverb (fluently) used instead of an adjective (fluent)

33: Accoridng to …  typo error

73: Virtual sensors are entities that indirectly measure … redundant words “are entities that

143: …concepts creates… improper form of the verb - plural versus singular

150: a survey most common … a preposition is missing (a survey of most common)

and many others.

English must be improved.

Author Response

We are grateful for your review. We have incorporated the comments and suggestions into our contribution. We attached a pdf file with highlighted changes. The description of our changes is in the table below.

Comments and suggestions

Our changes

The reviewed manuscript is a review paper that provides a taxonomy of sensors intended for intelligent transport systems (ITS). It offers a general overview of monitoring tasks in the road transport domain, with a specific focus on sensors designed for motion detection and object tracking. The categorization used allows for a comparison of the effectiveness of the mentioned kinds of sensors. The literature survey resulted in the design of a custom architecture, which was created and tested within the framework of a case study. Unlike other similar reviews, this paper considers physical and virtual sensors and data processing at different computing levels (cloud, foggy, edge). One of the strengths of this review is the design of a universal software-based virtual edge-enabled intelligent sensor system (EEISS) architecture.

The manuscript is clear, relevant for the Sensors journal field and presented well-structured – beginning with a general survey and taxonomy, comparison and concluding results of own research.
Thank you for the reviewer's statement.
However, the text is burdened with errors and grammatical inaccuracies. The English was checked and corrected. Changes are not highlighted because of their quantity.
Line 294: The statement “Ultrasonic sensors use sound waves transmitted at frequencies 25 and 50 kHz …” is too categorical and principally not true since there are ultrasonic sensors that operate in the range of 20 to 25 kHz as well. These sensors are typically used for applications that require longer detection ranges or the detection of larger objects (for example Murata MA40S4S - an ultrasonic sensor that operates at a frequency of 20 kHz and has a detection range of up to 5 meters; commonly used in applications such as parking assist systems, object detection, etc.) We corrected the Ultrasonic Sensor paragraph. Now, the description is more general. (lines 307-311)
The overview of sensors needed for the domain of ITS is not complete. For example, weather stations can also play a critical role by providing real-time data on weather conditions that can impact road safety and traffic flow. They are typically deployed along roadways or in strategic locations to monitor weather parameters such as temperature, humidity, wind speed, precipitation, and visibility. I admit that is not in the category of motion detection and object tracking, discussed in principle in the paper, but it could be mentioned on the side. We added information about weather sensors to ITS in lines 160-164.
Another missing sensors, which could belong to tracking and detection tasks and which should be mentioned and categorized are “high occupancy sensors” - they can help to manage and optimize the use of such called “High Occupancy Vehicle” (HOV) lanes, which are reserved for vehicles with multiple occupants (typically two or more). Again, they use a variety of technologies to detect the number of occupants in a vehicle. Common technologies include infrared sensors, microwave sensors, and video-based systems. The data collected by high occupancy sensors can be used to enforce HOV lane restrictions, monitor compliance with HOV lane regulations, and provide real-time information to drivers via variable message signs. Camera-based HOV approach can be easily included within a concept of the EEISS proposed by the authors. The information about HOV sensors was added to our study/review in lines 357-366, 503-511, and Table 3.
Cited references come from the period 1954 – 2023. They are relevant, most of them are recent publications (3-7 years old). There is only 1 self-citation – [75]. All references are cited in the text.
Recommendation: include other latest review-like sources related to intelligent transportation systems, their sensors and usage of IoT, especially:
https://www.mdpi.com/1424-8220/23/8/3880 (2023)
https://www.ijrti.org/papers/IJRTI2109012.pdf (2022)
https://ijettjournal.org/Volume-70/Issue-6/IJETT-V70I6P217.pdf (2022)
https://ieeexplore.ieee.org/document/9792638 (2022)
The citations were implemented to the contribution (Table 3, Section 2 in lines 65-67, and Section 5 in lines: 518-519).
The manuscript is scientifically sound. The case study with the proposed architecture of EEISS is appropriate to test the concept of a universal ITS monitoring solution. Thank you for the reviewer's statement.
Comments on the interpretation of Table 2: At first glance (from a terminological point of view), the classification of sensors used in a vehicle into in-vehicle and non-vehicle might be a bit confusing. An additional explanation/definition would be appropriate. If I understand the authors properly, in-vehicle sensors are built-in vehicle sensors while non-vehicle sensors are those additionally installed in a vehicle, e.g. by a user. We added definitions of in-vehicle and non-vehicle sensors in lines 341-344.
Recommendation: The third photo in Fig. 5 (farthest to the right) could be purposefully edited to increase its brightness and readability (unlike the photos given on the left, its resolution, so necessary for optical character recognition, is sufficient). All figures were redrawn to the vector graphics by the authors to increase the image quality, except Figure 5. Figure 5 has changed the brightness of the third photo on the right.
I agree with the presented conclusions, they seem to be consistent with the arguments presented within the literature survey. No ethical issues were identified in the document that needed to be addressed (especially in relation to Figure 5). Thank you for the reviewer's statement.
English style needs revision. To illustrate it a few samples of typical inaccuracies contained in the text: 2: The internet of things (IoT) ... should be with capitals (The Internet of Things (IoT)) 31: Organizing fluently and safe traffic ... an adverb (fluently) used instead of an adjective (fluent) 33: Accoridng to … typo error 73: Virtual sensors are entities that indirectly measure … redundant words “are entities that” 143: …concepts creates… improper form of the verb - plural versus singular 150: a survey most common … a preposition is missing (a survey of most common) and many others The English was checked and corrected. Changes are not highlighted because of their quantity.

Reviewer 2 Report

Manuscript ID sensors 2360675 Title: Review of IoT Sensor Systems Used for Monitoring the Road Infrastructure

Comment1: This work is Based on a survey of the literature with a particular emphasis on motion detection and object tracking techniques, this research 9 evaluated and classified the monitoring duties handled by intelligent transportation systems. based on 11The literature review of sensor systems used for 12 monitoring tasks in the intelligent transportation system was done in addition to the sensor and monitoring task classification. It was examined 13 accomplished their findings to measure, sense, or categorize the occurrences in the 14 monitoring tasks of the intelligent transportation system in this study. On the basis of motion detection and object tracking techniques 16 in intelligent transportation activities, the review's results were utilized to suggest the own design of the universal 15 sensor system for common monitoring jobs. A case study scenario was used to construct and evaluate the suggested architecture for the first 17 experimental outcomes. The authors can compare with another parameters case study

2. Figures Quality is very poor, add the high-Quality figures and must be draw by the authors only

3. Why Confusion matrix formulas are added into Table, these should be write using equation editor or using word editor only 

4. Write proper use of confusion matrix metrices properly , write proper equations

5. Figure name and Title should be below the figures 

6. Result figures also not Good, axis is not clear 

7. Add research more reference papers from year 2020 to 2023.

In Introduction 1 Paragraph1  there are 12 grammatical errors. 

Remove all errors from all the paragraph.

Author Response

We are grateful for your review. We have incorporated the comments and suggestions into our contribution. We attached a pdf file with highlighted changes. The description of our changes is in the table below.

Reviewer´s comments and suggestions Our changes
This work is Based on a survey of the literature with a particular emphasis on motion detection and object tracking techniques, this research evaluated and classified the monitoring duties handled by intelligent transportation systems based on the literature review of sensor systems used for monitoring tasks in the intelligent transportation system was done in addition to the sensor and monitoring task classification. It was examined accomplished their findings to measure, sense, or categorize the occurrences in the monitoring tasks of the intelligent transportation system in this study. On the basis of motion detection and object tracking techniques in intelligent transportation activities, the review's results were utilized to suggest the own design of the universal sensor system for common monitoring jobs. A case study scenario was used to construct and evaluate the suggested architecture for the first experimental outcomes. The authors can compare with another parameters case study. Thank you for the reviewer's statement.
Figures Quality is very poor, add the high-Quality figures and must be draw by the authors only All figures were redrawn to the vector graphics by the authors to increase the image quality, except Figure 5. Figure 5 has changed the brightness of the third photo on the right.
Why Confusion matrix formulas are added into Table, these should be write using equation editor or using word editor only. Write proper use of confusion matrix metrices properly , write proper equations We do not use any contingency tables, confusion matrixes, formulas, or equations in this paper.
Figure name and Title should be below the figures The titles in tables and figures were solved via a prepared MDPI latex template.
Result figures also not Good, axis is not clear All figures were redrawn to the vector graphics by the authors to increase the image quality, except Figure 5. Figure 5 has changed the brightness of the third photo on the right.
Add research more reference papers from year 2020 to 2023. We added 14 references from the year 2020 to 2023. All changes with new references are highlighted in the attached pdf file (lines: 65-67, 357-367, 503-511, 518-519, and Table 3).
In Introduction 1 Paragraph1 there are 12 grammatical errors Remove all errors from all the paragraph. The English was checked and corrected. Changes are not highlighted because of their quantity in the whole article.

Round 2

Reviewer 2 Report

All queries are replied. Good luck for work ahead